# OpenReview forum: "Out-of-Distribution Detection using Synthetic Data Generation"
_ICLR.cc/2025/Conference — Submitted to ICLR 2025_

### Official Review · Reviewer_Bv8Q · 2024-10-31

**Soundness:** 2
**Presentation:** 1
**Contribution:** 2
**Rating:** 3
**Confidence:** 4

**Summary:**

This paper proposes to use Language Models to generate synthetic OOD data to train OOD detectors.
Their experiments show that the synthetic OOD data can help improve the performance of the OOD detection.

**Strengths:**

1. This paper explores the possibility of using language models to help improve the performance of the  OOD detector.

2. The experiments consider various baselines and different OOD cases, such as near OOD and far OOD.

3. The author considers different tasks including toxicity detection, harm detection and RLHF reward modeling.

**Weaknesses:**

1. The overall writing and the formatting are poor. The arrangement of the figures all around and the main text always refers to figures in the appendix, which should be avoided (line 192, 212, 400, ). The logic between sections is not very clear. The author first showed the results and comparisons to baselines in section 3 without describing the method and setup first. Then the author refers to Table 1 (at page 2) at section 5.3 (page 7) which is not intuitive.

2. The results in Table 5 show that using real OOD data outperforms the synthetic OOD data in most of the cases. This doesn't support the claim of the author about the quality of their synthetic data, which was discussed in section 4.

**Questions:**

I'm not sure if it is a fair comparison if you compare the synthetic OOD setup with other baselines which are not trained on any OOD data.
I think the result in Table 5 is a more realistic one where you compare the synthetic data with the real one. This tells you how good the synthetic OOD data is. I see the point of the OOD scarcity issue if you consider everything as training data, then it would be hard to find real-world OOD data for training the detector. But it would be nice to have some more analysis about the quality of the synthetic as the result in Table 5 is contrary to the discussion in line 259-266, stating that synthetic OOD data is nearly as effective as real OOD data.

---

> ### Author Response · Authors · 2024-11-20
>
> Thank you for reviewing our paper. We have dedicated considerable time and effort to thoroughly address your concerns. While our responses may be detailed, we have made every effort to be concise while ensuring clarity and comprehensiveness. We look forward to further discussion.
>
> **1. The overall writing and the formatting are poor. The arrangement of the figures all around and the main text always refers to figures in the appendix, which should be avoided. The author first showed the results and comparisons to baselines in section 3 without describing the method and setup first.**
>
> Thank you for your feedback. We appreciate your observations and haved addressed them in the revised draft. Regarding the placement of figures to the appendix, we acknowledge that it may seem disruptive. However, due to space constraints, we placed some supplementary details in the appendix to keep the main text focused on the core content.
> **Regarding Section 3, presenting the results on selective classification before describing the method was a deliberate design choice to motivate the study**. By showing the importance of synthetic data for selective classification, our aim was to provide context for why our approach is a promising direction for OOD detectoin in the forthcoming sections. However, **as per your suggestion, we have now updated the revised draft by explaining our method prior to the selective classification experiemtns.**
>
> **2. The results in Table 5 show that using real OOD data outperforms the synthetic OOD data in most of the cases. This doesn't support the claim of the author about the quality of their synthetic data, which was discussed in section 4.**
>
> Thank you for your comment. **We would like to clarify that in Table 5 (Table 6 in the updated draft), real OOD data outperforms synthetic OOD data in only 6 out of 24 cases (3 metrics \* 8 experiments = 24 cases), not across the board**.
> It's important to note that the model trained on the CC/BT-GSM8K pair shows exceptional generalization to the CC/BT-MBPP test pair, achieving strong results for both InD datasets for all three metrics. This, along with the positive results presented in Table 1 (now Table 3 in the revised version), supports the quality of the synthetic data used.
> Furthermore, while a model trained on the CC/BT-MBPP pair does not perform as well on the CC/BT-GSM8K test pair in terms of FPR95, the InD accuracy is on par or even better. We explicitly mention in the paper that addressing this gap in FPR95 performance is part of our future work as we believe that improving prompt diversity will be key to closing this gap and enhancing the performance of synthetic data in such cases.
>
> Moreover, **per your feedback, we have revised our statement about the quality of the synthetic data, originally discussed in Section 4 (now Section 3 in the revised draft)**. We clarify that our synthetic data is *comparable* to real OOD data and may offer greater diversity, *sometimes* leading to better generalization than real data.

---

> ### Author Response · Authors · 2024-11-20
>
> **3. I'm not sure if it is a fair comparison if you compare the synthetic OOD setup with other baselines which are not trained on any OOD data. I think the result in Table 5 is a more realistic one where you compare the synthetic data with the real one. This tells you how good the synthetic OOD data is. I see the point of the OOD scarcity issue if you consider everything as training data, then it would be hard to find real-world OOD data for training the detector. But it would be nice to have some more analysis about the quality of the synthetic as the result in Table 5 is contrary to the discussion in line 259-266, stating that synthetic OOD data is nearly as effective as real OOD data.**
>
> Thank you for your thoughtful comment. Please note that the baselines (MSP, Energy, ReAct, DICE) we used are standard baselines in the OOD detection literature, both in the text and image domains. Methods that incorporate external OOD data, such as Du et al. (ICLR, 2024), Katz-Samuels et al. (ICML, 2022), and Hendrycks et al. (ICLR, 2019), also compare against these baselines. Additionally, the setup in Table 1 (now Table 3 in the revised version) follows the standard approach used in these works, but we go a step further by evaluating our method on cross-generalization performance in Table 5 (now Table 6 in the revised version).
>
> **Note, our method and the baselines have access to the *same real InD data*, thus making it a fair comparison. In practice, when OOD robustness is lacking, collecting appropriate real data can be time-consuming and resource-intensive. As Yang et al. (2024) highlight, "approaches impose a strong assumption on the availability of OOD training data, which can be infeasible in practice." Nonetheless, we still consider an ideal/oracle baseline (trained directly on real OOD data). In contrast, our synthetic data approach offers an immediate, practical solution that avoids this assumption.**
>
> Regarding the quality of synthetic OOD data, we believe that our results in Table 1 (now Table 3 in the revised version), particularly the perfect zero FPR95 on far-OOD tasks and the FPR95 closest to the ideal model on near-OOD tasks, demonstrate the high effectiveness of our approach. Moreover, the results in Table 5 (now Table 6 in the revised version), which present our method's performance in cross-generalization experiments—a less commonly explored setting in the literature—further reinforce the robustness of the synthetic data.
>
> That said, we do acknowledge the performance gap in the CC/BT-MBPP pair in Table 5 (now Table 6 in the revised version). As noted in the paper, improving this performance is part of our future work. We believe that enhancing prompt diversity and creativity will be key to addressing this gap and further improving synthetic data quality in such cases.
>
>
> In summary, while we recognize the challenges with synthetic data in certain scenarios, the overall results indicate that the synthetic OOD data used in our study is both effective and of high quality.
>
> **References:**
>
> Dan Hendrycks, Mantas Mazeika, and Thomas Dietterich. Deep anomaly detection with outlier
> exposure. ICLR 2019.
>
> Julian Katz-Samuels, Julia B Nakhleh, Robert Nowak, and Yixuan Li. Training ood detectors in their natural habitats. ICML 2022.
>
> Xuefeng Du, Zhen Fang, Ilias Diakonikolas, and Yixuan Li. How does unlabeled data provably help out-of-distribution detection? ICLR 2024.
>
> Yang, Jingkang, et al. "Generalized out-of-distribution detection: A survey." International Journal of Computer Vision (2024): 1-28.

---

### Official Review · Reviewer_Pn4T · 2024-11-02

**Soundness:** 2
**Presentation:** 2
**Contribution:** 3
**Rating:** 5
**Confidence:** 3

**Summary:**

This paper studies the problem of out-of-distribution detection by using LLMs to generate synthetic OOD data, which can then be used to train OOD detectors without needing existing OOD data.

Overall, I think this paper is ready for publication. The results reported are not clear and well-presented at the moment. More importantly, it is unclear whether the definition of OOD detection that is considered in this paper is relevant.

Recommendations for Improvement:
To improve the quality of the paper in future re-submissions, I would encourage the authors to define more clearly the OOD task and argue why the definition they use is useful. In particular, engaging with the distinction between cross-tasks OOD versus within-task OOD.

To better discuss the contribution, the paper should disentangle the synthetic OOD data generation process from modifications to the OOD detector itself. First show that the synthetic OOD data generator is useful independently of the OOD detector used. Then, show that the OOD detector with 3 classes is useful independently of the training data. Finally, sell the combination of the two.

**Strengths:**

The authors have performed extensive experimental testing, which could be useful if the results were presented more clearly.

**Weaknesses:**

Overly Simplistic Definition of OOD:
The OOD detection tasks considered in this study, even those termed “near-OOD,” seem simplistic and easy to solve. For instance, distinguishing between tasks like CC and GSM8k does not appear to be particularly challenging. It is not clear what is the real-world relevance or difficulty of these OOD tasks. It would be very surprising that no baseline method perform well on these tasks.
Additionally, for the few baselines reported on table 1, there seems to be an unusual amount of repetitions of numbers, which seems to point to mistakes in reporting.

Do we need Synthetic OOD Data:
Given the definition of OOD used here as “task vs. task” detection, it’s unclear why synthetic data generation is necessary. Instead of synthetic data, one could simply use any existing task/dataset as OOD data for this problem, there would be plenty of them already available without the need for generating need data. This should actually be a baseline to test the usefulness of the synthetic data generation: Instead of training OOD detector with synthetic OOD data, train with other existing data as OOD.

Generally, I feel that it would be more interesting to focus on within-task OOD detections where the distribution shift comes from shifts in the labels distribution shift, input properties, or mappings between the inputs and labels. This setup would be significantly harder, more relevant, and more likely to need synthetic OOD data.

Clarity of the Presentation: The current structure of the paper is disorganized, with key elements of the methodology, metrics, and datasets introduced only later in the paper, despite being referenced earlier. There is also some confusion about the different types of contribution between modifications to the OOD training pipeline ( generating synthetic OOD data) and modifications to the OOD detector itself (adding a third class instead of using a binary setup). These two modifications should be evaluated separately to clarify their individual contributions.

**Questions:**

Why are there some many repeated numbers in Table 1?
Why the tasks presented in Table 1 are not solved by simple heuristics (different tasks clearly ask different questions, it should be fairly easy to recognise them)?

---

> ### Author Response · Authors · 2024-11-20
>
> Thank you for reviewing our paper. We have dedicated considerable time and effort to thoroughly address your concerns. While our responses may be detailed, we have made every effort to be concise while ensuring clarity and comprehensiveness. We look forward to further discussion.
>
> **1. even those termed “near-OOD,” seem simplistic and easy to solve. For instance, distinguishing between tasks like CC and GSM8k does not appear to be particularly challenging. It is not clear what is the real-world relevance or difficulty of these OOD tasks. It would be very surprising that no baseline method perform well on these tasks. Why the tasks presented in Table 1 are not solved by simple heuristics (different tasks clearly ask different questions, it should be fairly easy to recognise them)?**
>
> **It seems like you're referring to 'far-OOD' tasks, not 'near-OOD', as the example you provided—distinguishing between tasks like CC and GSM8k—fits the far-OOD category**. The notion of far- and near-OOD is not new and has been prevalent in several previous works including  Liu et al. (2023); Yang et al. (2022); Winkens et al. (2020). For example, Liu et al. (2023) consider InD as a Sentiment Analysis task and Far-OOD as a Question Classification task.
>
> **Far-OOD detection is crucial, especially in real-world applications like systems that need to detect and handle tasks such as math or coding problems differently**. For instance, when a system encounters math or code problems, it should avoid applying certain types of processing, such as a harmful content aligner (e.g. another LLM), which might be useful for general text but would be unnecessary (and costly) for math or code tasks.
>
> **Regarding your concern about baseline methods (e.g. MSP, ReAct Energy, DICE), while it may seem surprising that baseline methods perform poorly on far-OOD tasks like CC versus GSM8k, this is actually expected.**
> It's important to note that these techniques, originally developed for image tasks, are widely used as a standard in the text domain. However, these methods often struggle when applied to text, due to the inherent challenges of language data, such as greater variability in input forms, semantics, and structure. Several previous studies have demonstrated that these baselines tend to yield very high False Positive Rates (FPR95) when applied to text datasets. For instance, when tested on the SST-2-IMDB as an InD-OOD pair, these methods produced FPR95 scores of 77.7, 79.1, 79.6, and even 100% on MSP, ReAct Energy, and DICE (as shown in Table 8 of Baran et al.'s 2023 ACL paper; note that on most InD-OOD pairs FPR95 is above 50). In contrast, our method yields surprisingly low FPR95 (e.g. a perfect zero on far-ood tasks and an FPR95 closest to the ideal model on near-OOD tasks, see Table 1 (now Table 3in the revised version) in our paper).
>
> Reference: Mateusz Baran, Joanna Baran, Mateusz Wójcik, Maciej Zieba, and Adam Gonczarek. Classical out-of-distribution detection methods benchmark in text classification tasks. ACL 2023.
>
> **2.  There seems to be an unusual amount of repetitions of numbers, which seems to point to mistakes in reporting. Why are there some many repeated numbers in Table 1?**
>
> **There are no mistakes in the results reported in Table 1 (now Table 3 in the revised version)**. We believe the repetition you're noticing refers to the InD accuracy, which is the same for the baseline methods—MSP, Energy, DICE, and ReAct. This is expected because all these methods use the same underlying model for prediction and only differ in how they perform OOD detection.
> We included a detailed explanation of how these baselines work in Appendix A, so it's possible that the reviewer inadvertently overlooked this part.
>
> Moreover, we also include the code for reproducing our experiments, including the implementation of these baselines. Therefore, **the results in Table 1 (now Table 3 in the revised version) can be easily verified by running the provided code.**

---

> ### Author Response · Authors · 2024-11-20
>
> **3a. Do we need Synthetic OOD Data: Given the definition of OOD used here as “task vs. task” detection, it’s unclear why synthetic data generation is necessary.  Instead of synthetic data, one could simply use any existing task/dataset as OOD data for this problem.**
>
> We thank the reviewer for the insightful comment.
> **Indeed, it is possible to use existing datasets as OOD data**, and this approach has been explored in several previous works, including Hendrycks et al. (NeurIPS 2018, ICLR 2019), Zhang et al. (WACV, 2023), and more recently by Du et al. (ICLR 2024) and Katz-Samuels et al. (ICML 2022).
> For example, Du et al. (ICLR, 2024), Katz-Samuels et al. (ICML 2022), and Hendrycks et al. (ICLR 2019) all use existing data to improve OOD detection. However, their approach relies on the assumption that such external data is both sufficiently available and representative of real-world OOD scenarios. In practice, real-world OOD inputs are highly diverse and unpredictable, making it difficult to curate datasets that capture all potential distribution shifts; **as Yang et al. (2024) highlight, "approaches impose a strong assumption on the availability of OOD training data, which can be infeasible in practice," practical constraints have led to a shift in recent research toward settings where real OOD data is either unavailable or significantly limited.**
>
> In contrast, **synthetic OOD data generation allows us to create more controlled and flexible test conditions**. By creating diverse synthetic data (see Figure 4 - now Figure 3 in the revised draft) that simulates various distribution shifts, we can train a more robust OOD detector, capable of approaching ideal performance (See Table 1 (now Table 3 in the revised version)).
>
> Moreover, **to clarify, the notion of "task vs. task" OOD detection is not new in the literature**, with several prior works like Du et al. (ICLR 2024), Katz-Samuels et al. (ICML 2022), and Hendrycks et al. (ICLR 2019) all addressing this approach to OOD detection.
>
> **3b. This should actually be a baseline to test.**
>
> Thank you for the suggestion. In fact, **we have already included a more competitive baseline in our experiments**, which we refer to as "Original (Ideal)". This baseline assumes the availability of the original OOD data for training the detector, without the need for synthetic data. It serves as a competitive benchmark, as **it is directly trained on the original OOD data**, rather than relying on outlier data filtered from a pool of OOD+InD data that wrongly identifies some InD samples as OOD as done by Du et al. (ICLR 2024), Katz-Samuels et al. (ICML 2022).
> As shown in Table 1 (now Table 3 in the revised version), our model performs comparably to this "Original (Ideal)" baseline, matching a perfect zero FPR95 on far-OOD data, and being closest to it on near-OOD data.
>
> **4. Generally, I feel that it would be more interesting to focus on within-task OOD detections.**
>
> Thank you for your suggestion. The results in our selective classification experiments (section Section 4 of the revised draft) already address within-task OOD detection and show substantially improved performance in this setup.
>
>
> **5. Clarity of the Presentation: confusion about the different types of contribution between modifications to the OOD training pipeline (generating synthetic OOD data) and modifications to the OOD detector itself (adding a third class instead of using a binary setup).**
>
> **Thank you for your feedback. We'll make sure to clarify this in the final version.** Specifically, we will include results using existing methods with our synthetic OOD data to provide a clearer comparison. Additionally, we want to emphasize that the 3-way design is simply a matter of convenience and not a design choice for the success of our method.
>
>
> **References:**
>
> Dan Hendrycks, Mantas Mazeika, Duncan Wilson, and Kevin Gimpel. Using trusted data to train
> deep networks on labels corrupted by severe noise. NeurIPS, 2018.
>
> Dan Hendrycks, Mantas Mazeika, and Thomas Dietterich. Deep anomaly detection with outlier
> exposure. ICLR 2019.
>
> Jingyang Zhang, Nathan Inkawhich, Randolph Linderman, Yiran Chen, and Hai Li. Mixture outlier
> exposure: Towards out-of-distribution detection in fine-grained environments. WACV 2023.
>
> Julian Katz-Samuels, Julia B Nakhleh, Robert Nowak, and Yixuan Li. Training ood detectors in their natural habitats. ICML 2022.
>
> Xuefeng Du, Zhen Fang, Ilias Diakonikolas, and Yixuan Li. How does unlabeled data provably help out-of-distribution detection? ICLR 2024.
>
> Yang, Jingkang, et al. "Generalized out-of-distribution detection: A survey." International Journal of Computer Vision (2024): 1-28.

---

> ### Author Response · Authors · 2024-11-24
> **Evaluating the Impact of Synthetic Data vs. Three-Way Model Design**
>
> We were able to conduct **new experiments to clarify that the three-way design is primarily a matter of convenience rather than a critical design choice for the success of our approach**. We conducted experiments on several InD-OOD pairs, including CC-GSM8k, CC-SST-2, and CC-ToxiGen, where we trained a binary model alongside our three-class model, ensuring both models were trained on an equal number of samples for consistency.
>
> The results in the Tables below indicate that the two models perform comparably across all key metrics, suggesting that the **primary performance improvement stems from our synthetic data generation pipeline rather than the choice of a three-way model design**.
> Lastly, note that the InD accuracy is the same as other baselines since we use the same classifier and only differ in how OOD is detected, wherein the baselines use the scoring method detailed in Appendix A while our method uses synthetic data.
>
> **Table for GSM8K:**
>
> | InD   | Method             | FPR95↓ | AUROC↑  | InD Acc↑ |
> |-------|--------------------|--------|---------|----------|
> | CC    | Synthetic (Ours, 3-way model)   | 0.0 | 100.0 | 92.97 |
> |       | Synthetic (Ours, binary model)  | 0.0 | 99.99 | 92.04 |
>
>
> **Table for SST-2:**
>
> | InD   | Method             | FPR95↓ | AUROC↑  | InD Acc↑ |
> |-------|--------------------|--------|---------|----------|
> | CC    | Synthetic (Ours, 3-way model)   | 10.16| 97.66 | 89.95 |
> |       | Synthetic (Ours, binary model)  | 8.13 | 97.97 | 92.04 |
>
> **Table for ToxiGen:**
>
> | InD   | Method             | FPR95↓ | AUROC↑  | InD Acc↑ |
> |-------|--------------------|--------|---------|----------|
> | CC    | Synthetic (Ours, 3-way model)   | 12.66 | 96.59 | 89.26 |
> |       | Synthetic (Ours, binary model)  | 14.47 | 96.37 | 92.04 |

---

> > ### Comment · Reviewer_Pn4T · 2024-11-25
> > **Thanks for detailed answer**
> >
> > I thank the authors for the detailed answer about their work.
> >
> > The rebuttal gave me more confidence about the intrinsic value of the work, however I still think that the paper deserve another round of improvements before being ready for publications. In particular, what came out from the review and answer of the authors is the need to better position the work within the field, e.g., regarding near-OOD, far-OOD, within task-OOD. This would be achieved by reframing the introduction to better delineate the scope of the paper. Overall, I think the contributions of the paper are valuable but should be better framed, presented and discussed more clearly.
> >
> > [I have updated my score to reflect these discussions]

---

> > > ### Author Response · Authors · 2024-11-25
> > >
> > > We sincerely appreciate the reviewer’s response to our rebuttal. Please note that we have already made significant revisions to the paper (e.g. revamping the related work section and adding new experiments and explanations). We promise to improving the introduction as suggested.
> > >
> > > However, we strongly believe that rejecting the paper as "marginally below the acceptance threshold" due to the introduction seems somewhat harsh, especially considering that these revisions can be easily addressed in time for the camera-ready version. We hope the reviewer will consider a fair evaluation and we are always open to discussion.

---

### Official Review · Reviewer_ENt4 · 2024-11-03

**Soundness:** 3
**Presentation:** 3
**Contribution:** 3
**Rating:** 6
**Confidence:** 2

**Summary:**

This paper proposes a novel framework for OOD detection, which leverages LLMs to generate synthetic data for the training data with external OOD data source. Upon experiments on nine InD-OOD dataset pairs, this method is shown to be effective and outperforms baselines.

**Strengths:**

1. This paper is really easy to follow, with proper figures and content analysis. Also, the methods proposed are very simple, and, as far as I know, new.

2. The selected datasets and metrics are proper to me.

**Weaknesses:**

In sec4 last paragraph, the authors stated that "our synthetic data is nearly as effective as real OOD data, and possibly more diverse, in representing OOD samples." with only showing the figures of visualization. I believe more statistical analysis is needed to make this claim valid.

**Questions:**

1. I am interested in the models used. In different sections of the experiments, the authors seemed to use different models, e.g., Llama3 70b for generating synthetic data, Llama2 13b for fine-tuning on the datasets tested, and Starling 7B for the RLHF model. Although they are all Llama based models, they are different versions with different parameter settings. Why did you use different settings rather than being consistent?

2. In prompting the model to generate synthetic data, did you use a fixed prompt template? Have you tried different prompt templates? Is it  necessary to test the robustness with different prompts? What kind of decoding strategy did you use?

---

> ### Author Response · Authors · 2024-11-20
>
> Thank you for reviewing our paper. We have dedicated considerable time and effort to thoroughly address your concerns. While our responses may be detailed, we have made every effort to be concise while ensuring clarity and comprehensiveness. We look forward to further discussion.
>
> **1. The authors seemed to use different models, e.g., Llama3 70b for generating synthetic data, Llama2 13b for fine-tuning on the datasets tested, and Starling 7B for the RLHF model. Why did you use different settings rather than being consistent?**
>
> The Llama3 70B variant was used for data generation because larger models tend to produce better, more coherent generations. However, we also conducted additional experiments using the Llama-3 8B-instruct to generate the synthetic data and evaluated its performance on several InD-OOD pairs, including CC-GSM8k, CC-SST-2, and CC-ToxiGen. The results for these pairs are shown in the tables below, with new results highlighted in **bold**:
>
> **Table for GSM8K:**
>
> | InD   | Method             | FPR95↓ | AUROC↑  | InD Acc↑ |
> |-------|--------------------|--------|---------|----------|
> | CC    | Original (Ideal)   | 0.00   | 100.00  | 93.85    |
> |       | MSP                | 100.00 | 41.11   | 92.04    |
> |       | Energy             | 96.36  | 54.81   | 92.04    |
> |       | ReAct              | 96.74  | 69.78   | 92.04    |
> |       | DICE               | 97.57  | 65.10   | 92.04    |
> |       | Synthetic (Ours-70B)   | 0.00 | 100.00 | 92.97 |
> |       | Synthetic (Ours-8B)   | **0.00** | **100.00** | **92.42** |
>
> **Table for SST-2:**
>
> | InD   | Method             | FPR95↓ | AUROC↑  | InD Acc↑ |
> |-------|--------------------|--------|---------|----------|
> | CC    | Original (Ideal)   | 0.055  | 99.99   | 92.60    |
> |       | MSP                | 92.31  | 54.27   | 92.04 |
> |       | Energy             | 70.35  | 73.25   | 92.04 |
> |       | ReAct              | 61.89  | 82.31   | 92.04 |
> |       | DICE               | 69.63  | 80.31   | 92.04 |
> |       | Synthetic (Ours-70B)   | 10.16 | 97.66 | 89.95    |
> |       | **Synthetic (Ours-8B)**  | **13.62** | **95.76** | **90.11** |
>
> **Table for ToxiGen:**
>
> | InD   | Method             | FPR95↓ | AUROC↑  | InD Acc↑ |
> |-------|--------------------|--------|---------|----------|
> | CC    | Original (Ideal)   | 4.79   | 98.67   | 89.68    |
> |       | MSP                | 92.77  | 65.80   | 92.04 |
> |       | Energy             | 84.89  | 68.74   | 92.04 |
> |       | ReAct              | 84.04  | 67.60   | 92.04 |
> |       | DICE               | 83.83  | 63.43   | 92.04 |
> |       | Synthetic (Ours-70B)   | 12.66 | 96.59 | 89.26    |
> |       | **Synthetic (Ours-8B)**  | **18.82** | **94.42** | **92.23** |
>
>
>
> As seen in the tables above, even a smaller model like Llama-3 8B-instruct is able to generate data capable of achieving perfect zero FPR95 on the far-OOD CC-GSM8k InD-OOD pair. Furthermore, on near-OOD datasets, its performance is second only to the Ideal baseline, showing that smaller models can still generate high-quality synthetic data for OOD detection tasks.
> **We have added these results in Table 3 of the updated paper along with explanations in Section 5.3. We plan to continue evaluating the remaining InD-OOD pairs and will update our results in the final version of the paper.**
>
>
> **For the detector models, we chose smaller 7B and 13B Llama variants because detector systems are meant to be simpler and computationally efficient**. Their primary function is to filter user inputs detected as OOD and avoid predictions on them. Using larger models would complicate the system unnecessarily and increase computational costs.
>
> For the RLHF experiment, **we used Starling-RM-7B-alpha because, unlike general Llama models, it is a pre-trained reward model specifically designed for the RLHF pipeline**. It is optimized to assign scores to model outputs, reducing the need for continuous human labeling.
> Moreover, we chose Starling-RM-7B-alpha in particular because, like many reward models, it excels in certain areas, such as achieving a 98.0% win rate in the Chat category on the RewardBench Leaderboard, but its performance drops to just 58.0% in the Reasoning category.
> **Our goal is to redesign the reward model so that it serves two purposes**: not only will it evaluate LLM responses with a score, but it will also classify those responses as either high-performing (InD) or low-performing (OOD), based on their win rate. The model will output two things: 1) a score, and 2) a classification label (InD or OOD). This dual-purpose approach enhances the RLHF pipeline by enabling practitioners to filter out responses where this reward model underperforms, ultimately helping to train a stronger and more reliable LLM.

---

> ### Author Response · Authors · 2024-11-20
>
> **2. In prompting the model to generate synthetic data, did you use a fixed prompt template? Have you tried different prompt templates? Is it necessary to test the robustness with different prompts? What kind of decoding strategy did you use? What kind of decoding strategy did you use?**
>
> In our preliminary experiments, we tried several prompt templates for the data synthesis process. We aimed to refine the prompts to achieve higher quality and more diverse results by manually inspecting a few generated samples. Examples of generated synthetic data and the final prompts are detailed in Tables 7-20.
>
> **We used a top-k decoding strategy for generating outputs**. At each step, the model considers only the top-k most probable tokens from the probability distribution predicted by the model, limiting the candidate set to the most relevant tokens.

---

### Official Review · Reviewer_GLNb · 2024-11-03

**Soundness:** 2
**Presentation:** 2
**Contribution:** 2
**Rating:** 5
**Confidence:** 3

**Summary:**

This paper explores the simple idea of generating synthetic OOD data for training OOD detectors. The core idea is to generate near-OOD and far-OOD data by prompting an LLM given the ID data. The synthetic data are generated with Llama-3 Instruct.

The authors ran experiments on three tasks: toxicity detection, harm detection, and reward modeling data classification. These experiments are done on Llama-2 13B and Starling-RM-7B-alpha.

The empirical results are generally positive across different datasets and experiment setups, although I have some questions and concerns as explained in later sections.

**Strengths:**

- Experiments are pretty comprehensive.
- Empirical results are generally positive.

**Weaknesses:**

I don't find this paper particularly well-recognized, and I had a hard time finding some relevant experiment details. Can I clarify:
1. Are all the baseline methods (MSP, Energy, DICE) trained with the original real data? And are the synthetic data the same size as the original OOD data?

2. You are using a 70B model to generate the synthetic data but using 13B or 7B data for the OOD detection task. In a way this is distillation? Have you analyzed the impact of the size of the synthetic data generation model? Would a 7B data be able to generate high-quality OOD data?

This is probably minor but I really don't like the way your wrote your related work section (first two paragraphs). Dumping a bunch of citations with minimal descriptions is not particularly useful.

**Questions:**

See above.

---

> ### Author Response · Authors · 2024-11-20
>
> Thank you for reviewing our paper. We have dedicated considerable time and effort to thoroughly address your concerns. While our responses may be detailed, we have made every effort to be concise while ensuring clarity and comprehensiveness. We look forward to further discussion.
>
> **1. Are all the baseline methods (MSP, Energy, DICE) trained with the original real data? And are the synthetic data the same size as the original OOD data?**
>
> The MSP, Energy, and DICE baselines are trained only on the in-distribution (InD) data and do not incorporate any out-of-distribution (OOD) data, neither original nor synthetic, during training. These methods are well-established in the OOD detection literature and follow a standard, widely accepted approach. As **detailed in Appendix A, these baselines utilize a $K$-class model trained solely on InD data to produce binary (i.e. OOD vs. InD) predictions using a scoring function and threshold**.
> Due to space limitations, we provided a detailed explanation in Appendix A, and it is possible the reviewer may have inadvertently overlooked these details.
>
> For the methods that use OOD data (i.e., Original (Ideal) and Synthetic (Ours)), **the size of the synthetic and original data is kept similar in our experiments**. However, it's important to note that synthetic data can be generated in large amounts, and our approach isn't limited by the volume of data. This flexibility allows us to improve the model’s performance with more data, if needed. We chose to keep the sizes of the synthetic and original data similar for consistency. Moreover, note that **using the real OOD data is an idealized baseline, which isn’t commonly used in OOD research**.
> Real-world OOD data can vary widely, and we often don’t have enough of it. Still, we think it's valuable to compare against this ideal baseline, as our results are closest to ideal, demonstrating the effectiveness of our approach.
> We apologize for not mentioning this earlier and have included it in the revised version of the paper.
>
> **2. You are using a 70B model to generate the synthetic data but using 13B or 7B data for the OOD detection task. In a way this is distillation? Have you analyzed the impact of the size of the synthetic data generation model? Would a 7B data be able to generate high-quality OOD data?**
>
> We conducted additional experiments to address your question regarding the use of smaller models for generating synthetic data. Specifically, we used Llama-3 8B-instruct to generate the data and evaluated its performance on several InD-OOD pairs, including CC-GSM8k, CC-SST-2, and CC-ToxiGen. The results are shown in the tables below, with new results highlighted in **bold**:
>
> **Table for GSM8K:**
>
> | InD   | Method             | FPR95↓ | AUROC↑  | InD Acc↑ |
> |-------|--------------------|--------|---------|----------|
> | CC    | Original (Ideal)   | 0.00   | 100.00  | 93.85    |
> |       | MSP                | 100.00 | 41.11   | 92.04    |
> |       | Energy             | 96.36  | 54.81   | 92.04    |
> |       | ReAct              | 96.74  | 69.78   | 92.04    |
> |       | DICE               | 97.57  | 65.10   | 92.04    |
> |       | Synthetic (Ours-70B)   | 0.00 | 100.00 | 92.97 |
> |       | Synthetic (Ours-8B)   | **0.00** | **100.00** | **92.42** |
>
> **Table for SST-2:**
>
> | InD   | Method             | FPR95↓ | AUROC↑  | InD Acc↑ |
> |-------|--------------------|--------|---------|----------|
> | CC    | Original (Ideal)   | 0.055  | 99.99   | 92.60    |
> |       | MSP                | 92.31  | 54.27   | 92.04 |
> |       | Energy             | 70.35  | 73.25   | 92.04 |
> |       | ReAct              | 61.89  | 82.31   | 92.04 |
> |       | DICE               | 69.63  | 80.31   | 92.04 |
> |       | Synthetic (Ours-70B)   | 10.16 | 97.66 | 89.95    |
> |       | **Synthetic (Ours-8B)**  | **13.62** | **95.76** | **90.11** |
>
> **Table for ToxiGen:**
>
> | InD   | Method             | FPR95↓ | AUROC↑  | InD Acc↑ |
> |-------|--------------------|--------|---------|----------|
> | CC    | Original (Ideal)   | 4.79   | 98.67   | 89.68    |
> |       | MSP                | 92.77  | 65.80   | 92.04 |
> |       | Energy             | 84.89  | 68.74   | 92.04 |
> |       | ReAct              | 84.04  | 67.60   | 92.04 |
> |       | DICE               | 83.83  | 63.43   | 92.04 |
> |       | Synthetic (Ours-70B)   | 12.66 | 96.59 | 89.26    |
> |       | **Synthetic (Ours-8B)**  | **18.82** | **94.42** | **92.23** |
>
> As seen in the tables above, even a smaller 8B model is able to generate data capable of achieving perfect zero FPR95 on the far-OOD CC-GSM8k InD-OOD pair. Furthermore, on near-OOD datasets, its performance is second only to the Ideal baseline, showing that smaller models can still generate high-quality synthetic data for OOD detection tasks.
>
> **We added these results in Table 3 of the updated paper along with explanations in Section 5.3. We plan to continue evaluating the remaining InD-OOD pairs and will update our results in the final version of the paper.**

---

> ### Author Response · Authors · 2024-11-20
>
> **3.  I really don't like the way your wrote your related work section (first two paragraphs)**
>
> Thank you for your feedback.  We have thoroughly revised the related work section to address your concerns and improved its clarity and strcuture in the revised version of the paper (changes highlighted in red).

---

### Meta-Review · Area_Chair_YydM · 2024-12-21

**Metareview:**

The paper proposes the use of synthetic OOD data to improve the OOD detection accuracy across a few benchmarks.

**Strengths:** Synthetic data is becoming extremely common and potentially valuable for pretraining, hence the detection of OOD seems relevant.

**Weaknesses:** I do not see why this framework is necessarily novel: is the OOD detection just the same as synthetic data detection. The examples provided in the appendix are not convincingly out-of-distribution compared to the original data. It is not clear how the generation process controlled for the “OOD”ness of the generated data.

**Reasoning for reject:** There were clarity issues with the presentation of the work as well as the results. Moreover, the work was not novel, the synthetic data generation did not account for the OODness of the data.

**Additional Comments On Reviewer Discussion:**

It is clear that none of the reviewers found the original paper convincing in their method description, comparison with related work (of which there is a lot), empirical details (for instance, reviewers found there to be many apples-to-oranges comparisons with some baselines being trained on real data and only the authors’ framework being trained on synthetic data). While there was not a lot of reviewer discussion for this paper, it is possible that the reviewers did not find the responses by the authors compelling enough to change their assessments. Moreover the authors’ response was not succinct and too lengthy for any reviewer to meaningfully engage with it.

---

### Decision · Program_Chairs · 2025-01-22

Reject